# High expression of *Helicobacter pylori* VapD in both the intracellular environment and biopsies from gastric patients with severity

Rosario Morales-Espinosa[1]*, Gabriela Delgado[1], Luis-Roberto Serrano[1], Elizabeth Castillo[1], Carlos A. Santiago[1], Rigoberto Hernández-Castro[2], Alberto Gonzalez-Pedraza[1], Jose L. Mendez[1], Luis F. Mundo-Gallardo[3], Joaquín Manzo-Merino[4], Sergio Ayala[4], Alejandro Cravioto[1]

**1** Facultad de Medicina, Departamento de Microbiología y Parasitología, Universidad Nacional Autónoma de México, Mexico City, Mexico, **2** Departamento de Ecología de Agentes Patógenos, Hospital Manuel Gea González, Mexico City, Mexico, **3** Gastroenterología, Hospital Angeles del Pedregal, Mexico City, Mexico, **4** Cátedras CONACyT-Instituto Nacional de Cancerología, Mexico City, Mexico

* marosari@unam.mx

## Abstract

*Helicobacter pylori* is a Gram-negative bacterium that causes chronic atrophic gastritis and peptic ulcers and it has been associated with the development of gastric adenocarcinoma and mucosa-associated lymphoid tissue (MALT). One of the more remarkable characteristics of *H. pylori* is its ability to survive in the hostile environment of the stomach. *H. pylori* regulates the expression of specific sets of genes allowing it to survive high acidity levels and nutrient scarcity. In the present study, we determined the expression of virulence associated protein D (VapD) of *H. pylori* inside adenocarcinoma gastric (AGS) cells and in gastric biopsies. Using qRT-PCR, VapD expression was quantified in intracellular *H. pylori*-AGS cell cultures at different time points and in gastric mucosa biopsies from patients suffering from chronic atrophic gastritis, follicular gastritis, peptic ulcers, gastritis precancerous intestinal metaplasia and adenocarcinoma. Our results show that *vapD* of *H. pylori* presented high transcription levels inside AGS cells, which increased up to two-fold above basal values across all assays over time. Inside AGS cells, *H. pylori* acquired a coccoid form that is metabolically active in expressing VapD as a protection mechanism, thereby maintaining its permanence in a viable non-cultivable state. VapD of *H. pylori* was expressed in all gastric biopsies, however, higher expression levels (p = 0.029) were observed in gastric antrum biopsies from patients with follicular gastritis. The highest VapD expression levels were found in both antrum and corpus gastric biopsies from older patients (>57 years old). We observed that VapD in *H. pylori* is a protein that is only produced in response to interactions with eukaryotic cells. Our results suggest that VapD contributes to the persistence of *H. pylori* inside the gastric epithelial cells, protecting the microorganism from the intracellular environment, reducing its growth rate, enabling long-term infection and treatment resistance.

**Data Availability Statement:** All relevant data are within the manuscript and its supporting information files.

**Funding:** RME received: DGAPA-PAPIIT grant IN213816. RME received: CONACYT grant CB-255574.

**Competing interests:** The authors have declared that no competing interests exist.

## Introduction

*Helicobacter pylori* infection is the most common chronic infection around the world, since different epidemiological studies have shown that approximately 50% of the human population is infected with the bacterium [1]. However, the development of gastric disease is the result of many years of persistent *H. pylori* infection (colonization) in the gastric mucosa, where it alters the production of gastric hormones, affecting the gastric physiology and producing structural damage of the gastric cells. *H. pylori* has been associated with different gastric pathologies, such as chronic gastritis, gastric atrophy, follicular gastritis, duodenal ulcers, gastric ulcers, intestinal metaplasia, dysplasia, gastric adenocarcinoma (GAC) and gastric lymphoma MALT [2]. *H. pylori* colonization induces an inflammatory response; however, the microorganism uses mechanisms to protect itself from the immune response and from being eliminated from the gastric mucosa, thereby allowing long-time persistence. Various virulence factors are associated with gastric pathologies, such as a vacuolizing cytotoxin (VacA), the cytotoxin-associated gene A (CagA), the *cag* pathogenicity island and outer-membrane proteins (OMPLA, BabA, OipA) [3]. However, the prevalence of these virulence factors is diverse among *H. pylori* strains and between isolates from different geographic areas and ethnic groups [4]. This variability is due to the fact that *H. pylori* is a microorganism with high genetic diversity at both the gene and chromosomal levels. For example, although the *vacA* gene, is a structural gene that is present in all *H. pylori* strains, there is only a 65% nucleotide sequence identity between the *vacA* gene of cytotoxin-negative and cytotoxin-positive strains, and approximately 50% of strains contain an active toxin for inducing vacuolation of epithelial cells [5]. The *cag* pathogenicity island (*cag*-PAI) also has a variable structure and a different chromosomal arrangement; in some strains it presents as a single uninterrupted unit, or can be divided into two regions by IS605 (*cag*I and *cag*II), be split into *cag*I and *cag*II by a large segment of chromosome, or it can consist of partially deleted segments of *cag*I, *cag*II or both [6]. Another distinctive hallmark is the presence of strain-specific genes [7,8], which are found within plasticity zones where the highest diversity among *H. pylori* strains can be seen.

In addition to *vacA* and *cag*-PAI, there is a specific genetic locus, which is only carried by a proportion (from 36% to 61%) of strains. This genetic locus is an ORF related to virulence-associated protein D (*vapD*) and located 3.5 kb downstream from the *vacA* gene in *H. pylori* strain 60190 [9]. However, this ORF is located in a different place in the strain 26695 (HP0315), while it is truncated in the strain J99 [10]. *vapD* has been found in other microorganisms of different phyla and although its function is unknown, it has been attributed to the survival of *Rhodococcus equi*, and *Haemophilus influenzae* into macrophages and epithelial cells [11–16].

One strategy of some microorganisms to persist in an intracellular niche is to avoid the phagolysosome fusion. It is well documented that some strains of *H. pylori* invade the gastric mucosa cells and can persist for an undetermined time. This phenomenon has clinical implications, which can lead to chronic and/or recurring infection, as well as to treatment resistance.

The molecular mechanisms for which *H. pylori* can invade, replicate and persist in epithelial cells are not well known. However, these processes include at least four steps: a) the quick mobility of *H. pylori* from the lumen to the gastric mucosa; b) the adherence of *H. pylori* to the surface of gastric mucosa epithelial cells via outer membrane proteins; c) cellular invasion of *H. pylori* and finally, d) envelopment of *H. pylori* by double-layer membrane vesicles following invasion. However, whether *H. pylori* persists inside the cells for a long time or is killed outside the cells by lysosomal killing mechanisms 24 hours after invasion remains a controversial topic [4,17]. Studies have reported that *H. pylori* can survive inside phagocytic cells, inhibiting phagosome maturation by urease-derived ammonia pathways and thereafter, enhancing the activity of VacA secreted by intraphagosomal bacteria [18]. Results of a recent study suggest that *H.*

*pylori* uses a similar strategy to survive inside gastric epithelial cells. Nevertheless, only a small portion of *H. pylori* strains are intracellular. Type I *H. pylori* strains use their virulence factors to induce a chronic neutrophil-rich inflammatory response, in which factors, such as CagA and VacA, participate in the adhesion and internalization of the bacteria within phagocytic and epithelial cells [18]. Meanwhile, type II strains induce much less inflammation and in turn, a markedly reduced phagocytic influx, hence intracellular survival is not essential for type II *H. pylori* [18].

*Rhodococcus equi* is a Gram-positive bacterium and a facultative intracellular pathogen, which causes pneumonia in young foals. *R. equi* has a virulence plasmid, where a cassette of genes (*vapA-vapH*) are present, when the microorganism is subjected to oxidative or acidic stress, *vapA*, *vapD* and *vapG* genes are highly expressed. *R. equi* is highly resistant to hydrogen peroxide, which may affect the maturation process of the phagosome within macrophages, and in contrast to VapA and VapG, the VapD expression may be highly induced inside the macrophage [13,14]. Non-typeable *Haemophilus influenzae* (NTHi) is a common commensal of the upper respiratory tract. As well as being a significant cause of respiratory tract infections in humans, it also penetrates human respiratory cells (epithelial cells and macrophages). During infections of otitis media caused by NTHi, various stress stimuli, such as nutrient deprivation, antibiotics, and reactive-oxygen species encountered by the bacteria, may result in the activation of *vapD*, which forms part of the toxin-antitoxin module (*vapXD*), leading to the arrested growth of the bacteria in epithelial cells. *vapD* mutations have shown reduced NTHi persistence in a chinchilla model for otitis media [16]. Although, the *vapD* gene has been described in many other microorganisms of different gender and phyla, its specific role remains unclear. Kwon et al., [19] determined the structural and biochemical characteristics of *H. pylori* VapD and found that this protein displayed a purine-specific endoribonuclease activity, which later showed to be structurally related to the Cas2 proteins, but unlike *H. influenzae vapD* gene, in *H. pylori* the *vapD* gene, did not belong to a TA system [19].

Studies have shown the expression of virulence factors, transcriptional regulators and different proteins of *Helicobacter pylori* under different developmental conditions [20,21]. However, there are no previous studies documenting the expression of VapD in *H. pylori* at the intracellular environmental level nor in gastric biopsies from patients with different gastric pathologies. In the present work, we revealed that *vapD* is expressed in the intracellular environment and in gastric mucosa cells (*in vivo*); *vapD* presents high expression levels in severe gastric pathologies, which varied between patients of different ages, suggesting that *vapD* is participating in the chronicity of gastric infection.

## Materials and methods

### *H. pylori*-AGS cells co-culture

Human gastric adenocarcinoma cell lines (AGS [ATCC® CRL-1739]) were grown in Dulbecco´s Modified Eagle Medium (DMEM, Life Technologies®), supplemented with 5% fetal bovine serum (FBS, Corning Costar) and antibiotics. The AGS cells were incubated in 5% $CO_2$ atmosphere until 80% confluency was reached and then were distributed in 12 well plates. *H. pylori* strain 26695 (ATCC 700392) was grown in blood agar plates (BAP) supplemented with 5% fetal bovine serum in microaerobic conditions for 48 hours. AGS cells ($3.2X10^5$) with fresh DMEM were inoculated with *H. pylori* strain 26695 (*vapD* +) to a $3.2X10^7$ concentration of *H. pylori*. All of the resulting co-cultures were incubated in a $CO_2$ atmosphere for 6 hours (t0). Following incubation, each co-culture was inoculated with 200 μg/ml of gentamycin and incubated for an additional time of 6 hours (t1). After the second incubation, DMEM was removed from intracellular *H. pylori*-AGS cells (cellular package), thereby discarding any

microorganism that did not enter the AGS cells. The cellular packages were supplemented with fresh DMEM and maintained for 108 hours, with the medium being changed every 48 hours with fresh DMEM until the total number of hours established for the assay was reached. Each cellular package was treated separately at different time points [6-h (t0), 12-h (t1), 24-h (t2), 36-h (t3), 48-h (t4), 60-h (t5), 72-h (t6), 84-h (t-7), 96-h (t8) and 108-h (t9)]. At each time point, the cellular package (intracellular *H. pylori*—AGS cells) was separated from the DMEM. The cellular package was treated with 0.1% saponin in 1 ml PBS for 15 min. Saponin was used to permeabilize AGS cell membranes by penetrating the cholesterol monolayer and forming holes or pits allowing *H. pylori* to exit from the AGS cell. 50 μl of each sample (cellular package and DMEM) were plated on BAP supplemented with 5% fetal bovine serum and incubated in microaerobic conditions for 48 hours to determine the colonies forming unit (CFU) count. The results obtained from DMEM and the cellular package at time t1 (12 hours post-infection and treated with gentamycin) were taken as negative and positive controls for *H. pylori* growth and invasion, respectively.

In addition, *H. pylori* invasion was confirmed by Immunofluorescence and Western blot assays. The *vapD* transcription levels were obtained by qRT-PCR, confirming that *H. pylori* maintained its metabolic activity into intracellular niche. All assays were performed in triplicate at each time point.

## Immunofluorescence assay

Each cellular package was immunostained. The AGS cells were fixed in 3.7% formaldehyde and PBS at room temperature for 30 min, and later washed three times with PBS. The cells were permeabilized by incubating in 0.25% (vol/vol) Triton X-100 in PBS for 20 min and then blocked for 30 min with 5% milk in PBS (vol/vol), at room temperature. Following washing three more times, an antibody mixture: containing mouse anti-*H. pylori* antibodies (Biocare Medical®) at 1:500 dilution and rabbit anti-β-tubulin antibody (Biocare Medical®) at 1:500 dilution was added. The AGS cells with antibody mixture were incubated at 4˚C overnight. Following extensive washing, the mixture was incubated with Alexa 568 marked anti-mouse secondary antibody (Invitrogen®) and Alexa 488 marked anti-rabbit secondary antibody (Invitrogen®) at 1:1000 dilution at 4˚C for a further 2 hours. Finally, we used DAPI (4´, 6-diamino-2-phenilindol) stain to visualize the AGS cell nucleus. The samples were observed using confocal microscopy.

## Western blot

Total protein extraction was made from the cellular package and its DMEM (negative control of *H. pylori* presence) taken at different time points (6, 12, 24, 36, 48, 60, 72, 84, 96 and 108 hours). Intracellular *H. pylori*-AGS cell monolayers were washed three times with PBS and lysed by adding RIPA buffer 1X (Radio Immuno Precipitation Assay. BIOMAN Scientific CO., LTD) with a protease inhibitor (Complete, Rocher®) according to the manufacturer's instructions. Protein concentration was determined by DC Protein Assay (BIO RAD) following the manufacturer's instructions. Samples were loaded onto 10% SDS-PAGE gel and transferred to a 0.45μm PVDF *immobilion* Millipore® membrane. Western blotting was performed using mouse anti-*H. pylori* antibody (Biocare Medical®) at 1:500 dilution and anti-mouse secondary antibody (*Jackson Immuno Research*). The membrane was revealed using an EZ ECL kit (Biological industries®). Finally, the membrane was exposed on radiographic plates and revealed (Kodak®).

## Total RNA extraction and detection of *vapD* and *GAPDH* genes' transcription levels by qRT-PCR from intracellular *H. pylori*-AGS cells

Total RNA was isolated from the cellular package (intracellular *H. pylori*-AGS cells) at different time points using Hybrid-R™ (GeneAll Biotechnology Co., Ltd) according to the manufacturer´s instructions. The cellular package was harvested in 500 µl RiboEx™. The sample was homogenized using a 0.9 mm needle (20 gauge) and incubated at room temperature for 5 min. Following incubation 200 µl of chloroform were added and the sample was vigorously shaken for 15 sec and incubated at room temperature for 2 min. The sample was then centrifuged at 12,000 x g at 4˚C for 15 min and the aqueous phase was transferred to a fresh tube. Equal volumes of RBI buffer were added to the sample and thoroughly mixed by inversion. The mixture was transferred to a mini column and centrifuged at 10,000 x g for 30 sec. The mini column was then washed twice, firstly with 500 µl SWI buffer and then with 500 µl RNW buffer. RNA was eluted from the column by adding 40 µl nuclease-free water followed by centrifugation at 10,000 x g for 1 min. Spectrophotometrical analysis was used to determine total RNA concentration using NanoDrop 2000 (Thermo Scientific®).

Total RNA (1 µg) was reverse transcribed using a QuantiTect® Reverse Transcription Kit (QIAGEN®) according to the manufacturer´s instructions. The resulting cDNA was stored at -80˚C. *vapD* and GAPDH mRNA levels were determined using real-time PCR a Step one plus Real-Time PCR System (Applies Biosystems®). cDNA was denatured at 94˚C for 2 min and then subjected to 30 cycles of annealing at 94˚C for 15 s and extension at 60˚C for 1 min. The *vapD* gene was amplified using specific primers, previously documented by Cao and Cover [9]. A (TaqMan) FAM-AGAGCGTTTAAGGTAGAGGACTTTAGCGA probe, which was designed in our laboratory, was also used to detect *vapD*. For GAPDH detection, we used a Real-Time PCR TaqMan® by Life Technologies GAPDH control. The RT-PCR mix had a total volume of 25 µl using iQ™ Multiplex Powermix (BIO-RAD). Transcription levels of GAPDH were used as normalization values and as endogenous controls. The *vapD* transcription level at t1 (12 hours post-infection) was used as our calibrator and reference for 100% expression of the *H. pylori vapD* gene in the intracellular environment. Relative *vapD* gene expression was calculated using the $2^{-\Delta\Delta Ct}$ method [22]. Each assay was performed in triplicate.

## Clinical sample

A total of 82 gastric biopsy samples (41 antrum and 41 corpus) were taken from 41 patients: 23 women and 18 men whose ages ranged from 18 to 92 years with a median of 74 years. The patients were being treated in three different hospitals, namely, the gastroscopy service at the Manuel Gea González General Hospital and the Hospital Angeles del Pedregal both of which are located in Mexico City, and the gastro-endoscopy service at a private clinic located in Puerto Escondido, Oaxaca, Mexico. The criteria used to select patients were a previous clinical diagnosis of functional dyspepsia and gastric morphological alteration, at the time of endoscopy. All patients had received previous diagnosis of a gastric pathology associated with *H. pylori*: cancer (four), peptic ulcer (nine), follicular gastritis (six), chronic atrophic gastritis (nineteen); and three patients with gastric pathology, which was not determined. Written, informed consent was obtained from all patients.

Additionally, RNA from *H. pylori* 8822 (ATCC 51932; this strain is *vapD* negative) was used as a negative control for transcription.

The endoscopies involved two biopsy samples being taken from adjacent areas of the gastric antrum and gastric corpus; the samples were immediately stored in RNA-later medium (Ambion) and kept at 4˚C until RNA extraction.

### Ethics statement

The protocol was approved by the ethics commission from each hospital and the Faculty of Medicine with the 055/2014 number at the Universidad Nacional Autónoma de México (UNAM).

### RNA extraction from gastric biopsy

Tissue samples were transferred from RNA-later medium to new tubes and lysed in tissue grinder with 500 μl of RiboEx™, using Hybrid-R™ (GeneAll Biotechnology Co., Ltd) according to the manufacturer´s instructions. The RNA extraction protocol is described in the section above.

### Detection of *vapD, vacA and GAPDH* genes' transcription levels by RT-PCR from gastric biopsy

Detection of transcription levels of the *vapD* and GAPDH genes were performed as described above. The *vacA* gene was used as an internal control of *H. pylori* gene transcription since *vacA* is a constitutive gene of *H. pylori* which is always transcribed, independent of the allele presented. Specific primers for qRT-PCR amplification and detection of the *vacA* gene were designed in our lab: *vacA* forward: ATGGAAATACAACAAACACACC; *vacA* reverse: CCAACA ATGGCTGGAATGA and probe VIC-ACTTTGTTGCGGTGTGATGCTGAC. All samples were produced in triplicate.

### Statistical analysis

Comparisons were performed using one-way analysis of variance (ANOVA) and subsequent *t*-tests. Additionally, we applied U of Mann Whitney and Kruskall Wallis tests were applied.

## Results

We performed co-cultures of *H. pylori*-AGS cells and incubated over different time points (6-h (t0) 12-h (t1), 24-h (t2), 36-h (t3), 48-h (t4), 60-h (t5), 72-h (t6), 84-h (t-7), 96-h (t8) and 108-h (t9)]. After the first 6 hours of incubation, gentamycin was added, leaving the antibiotic to work for an additional time of 6 hours (t1, 12 hours post-infection). After each time point, the cellular package (intracellular-*H. pylori*- AGS cells) was separated from the DMEM and both were analyzed independently to determine the CFU count on blood agar plates. In addition, Western blot was performed to detect a specific heat shock protein of *H. pylori* with a molecular size of 70 kDa. The cellular package was treated with saponin to allow releasing of *H. pylori* from AGS cells. At each 12 hours interval, intracellular viability and persistence of *H. pylori* inside AGS cells were determined. In addition, the *vapD* transcription levels of *H. pylori* in the intracellular environment were measured using qRT-PCR. All assays were performed in triplicate.

To determine the viability of *H. pylori* inside AGS cells, the CFU counts from all samples at different time points were obtained. 50 μl of DMEM (negative control) and cellular package (pre-treatment and post-treatment with gentamycin) treated with saponin were plated on BAP supplemented with 5% fetal bovine serum and incubated in microaerobic conditions for 48 hours. At time t0 (co-culture pre-gentamycin treatment) were recuperated a $1.09 \times 10^6$ concentration of intracellular *H. pylori* and a $1.36 \times 10^6$ concentration of extracellular *H. pylori* (from DMEM). From the co-cultures at t1 to t9 (post-gentamycin treatment), *H. pylori* was obtained from the cellular package only (treated with saponin) but none was retrieved from DMEM, which confirms that *H. pylori* invaded the AGS cells and were eliminated from DMEM. From

the co-cultures at t1 to t4 (approximately 48 hours post-infection), the number of CFUs counted was constant with approximately 485 CFUs at t1, 433 CFUs at t2, 596 CFUs at t3 and 537 CFUs at t4. At time t5 (60 hours), there was an increase in the CFU count (1300), suggesting that *H. pylori* had been replicated within the AGS cells. At t6 there was a decrease in the CFU count (482 CFUs), and at t7, the count decreased notably and continued to decrease up to only obtained 12 CFUs at t9. Despite the significant reduction in CFU counts from t5 to t9, the immunofluorescence assay confirmed that *H. pylori* remained viable within AGS cells, and *vapD* transcription levels were detected by qRT-PCR.

To confirm that *H. pylori* was present only in the intracellular environment, we extracted total protein from both the cellular package and DMEM at the different time points. We performed a Western blot using mouse anti-*H. pylori* antibody, which detected a 70 kDa heat shock specific protein of *H. pylori*. In addition, we detected GAPDH using rabbit anti-GAPDH antibody as load control. Both the cellular package and DMEM samples at t0 (pre-gentamycin treatment) gave a positive signal with the mouse anti-*H. pylori* antibody, confirming the presence of *H. pylori* in both the intracellular (within AGS cells) and extracellular environments (DMEM). Nevertheless, from t1 to t9 time points (post-gentamycin and saponin treatment), detection of the heat shock protein using mouse anti-*H. pylori* antibody proved positive in samples from the cellular package but not from DMEM, indicating the exclusive presence of *H. pylori* in the intracellular environment (Fig 1).

In order to identify the location of *H. pylori* within the AGS cells, we performed immunofluorescence assays in all cell packages that had been infected with *H. pylori* and monitored them by confocal microscopy at different time points (from 6 hours up to 108 hours post-infection). Each cellular package was immunostained using an antibody mixture: mouse anti-*H. pylori* antibody (red), which was visualized using the 561 nm laser line of the Leica TCS SP5 confocal microscope at full power; rabbit anti-β tubulin antibody (green), which was visualized using the 488 nm laser line of the Leica TCS SP5 confocal microscope at full power; and DAPI (blue) to visualize the AGS cell's nucleus using the UV351-364 nm laser line of the Leica TCS SP5 confocal microscope at full power. Fluorescence and differential interference contrast images were obtained (Fig 2a). A three-dimensional image was created using the LAS AF software. The extracellular and intracellular *H. pylori* locations were visible in the first 6 hours post-infection (t0, without gentamycin). After 12 hours (t1, post-gentamycin treatment), we observed that *H. pylori* began to acquire different morphological forms such as bacillary, spiral and coccoid shapes within the AGS cells. The majority of the bacteria were localized near the

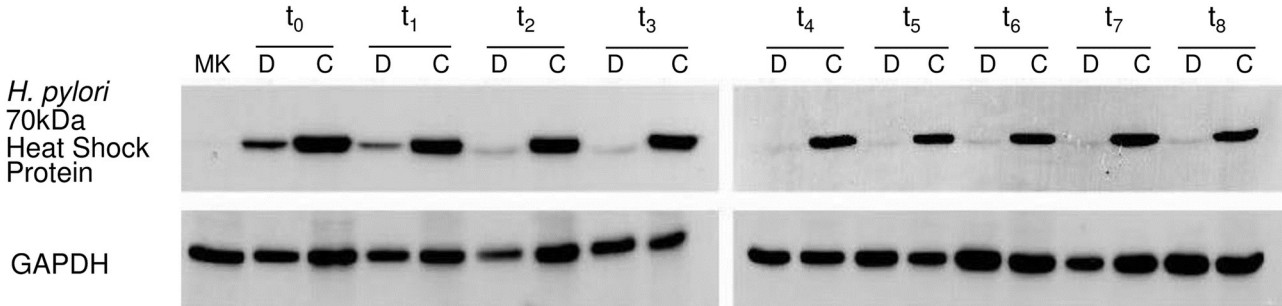

**Fig 1. Detection of a *H pylori* protein in the intracellular and extracellular environment.** A 70 kDa. specific heat shock protein was detected by Western blot using mouse anti-*H. pylori* antibody. Total protein extraction was made from the cellular package and its DMEM (negative control of *H. pylori* presence) taken at different time points: t0 (6 h of incubation, without gentamycin treatment), t1 (12-hours post-infection and treated with gentamycin), t2 (24-h), t3 (36-h), t4 (48-h), t5 (60-h), t6 (72-h), t7 (84-h) and t8 (96-h). D: Dulbecco´s Modified Eagle Medium; C: cellular package.

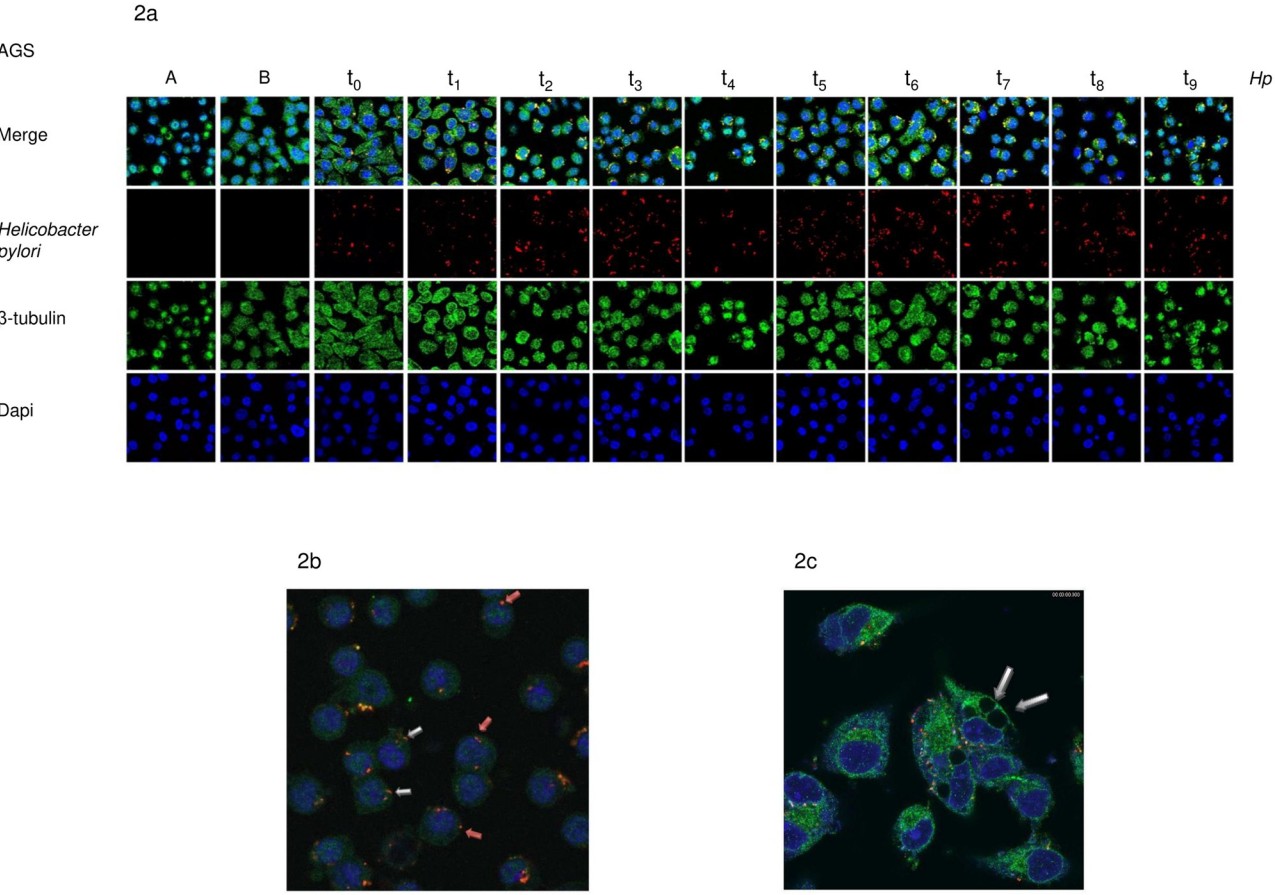

**Fig 2. Observation of the location of *H. pylori* within the AGS cells for immunofluorescence assays and confocal microscopy.** 2a. Images of noninfected AGS control cells are shown in the first two columns. The first row shows the different locations of *H. pylori* within AGS cells at different time points: t0 (6 hours post-infection and pre-gentamycin treatment); t1 (12-h [hours]); t2 (24-h); t3 (36-h); t4 (48-h); t5 (60-h); t6 (72-h); t7 (84-h); t8 (96-h) and t9 (108-h) post-infection and post-gentamycin treatment using a Leica TCS SP5 confocal microscope. In the second row, *Helicobacter pylori* is visualized using mouse anti-*H. pylori* antibody (red), which was visualized using the 561nm laser line. In third row, β-tubulin is visualized using rabbit anti-β tubulin antibody (green), which is observed using the 488 nm laser line. In the fourth row, the nucleus of AGS cells is visualized using DAPI (blue) and a UV351-364 nm laser line. b and c. Amplified images of AGS cells infected with intracellular *H. pylori*. In 2b, the coccoid form and bacillary form of *H. pylori* can be observed inside the AGS cells and a large reduction in the AGS cell cytoplasm is seen. In 2c, the formation of vacuoles in the cytoplasmic space of the AGS cells are observed but we could not visualize *H. pylori* inside the vacuoles.

nucleus, and a large reduction in the AGS cell cytoplasm was observed (Fig 2b). At 24 hours (t2), the formation of vacuoles in the cytoplasmic space of the AGS cells began to be observed (Fig 2c), although *H. pylori* could not be seen inside the large vacuoles. Throughout the assay, the bacillary form persisted although in lesser number, while the coccoid form of *H. pylori* increased in number (Fig 2a), as seen in the low CFU count. We now know that the conversion of *H. pylori* from bacillary to coccoid form can be induced when the bacterium is subjected to stress and this change in form coincides with it acquiring a viable non-cultivable state.

In order to determine if the *vapD* gene is transcribed inside gastric cells, and to corroborate viability of *H. pylori* within AGS cells, we used qRT-PCR to measure the *vapD* gene transcription levels of *H. pylori* strain 26695 in the intracellular environment at different time points. In addition, we measured the *vapD* gene transcription level of *H. pylori* strain 26695 grown in LB medium. The *vapD* transcription level of *H. pylori* at 6 hours (t0) (post-infection without gentamycin) showed a -ΔΔCt 1.8966 value. This value corresponded to VapD expression of *H.*

*pylori* present in both the extracellular and intracellular environments. The transcription level obtained at 12 hours (t1) post-infection, was taken as our reference value, which was assigned a value of 1.0. All the values obtained at the different time points were normalized against this reference value. We also took this value (1.0) as our reference value for 100% *H. pylori* VapD expression in the intracellular environment. Fig 3 details the *vapD* transcription levels at different time points (from t0 to t9) and shows that the VapD expression increased two-fold (from 1.460585 to 2.012828) in the first 36 (t3) hours of intracellular *H. pylori* infection; at 48 hours (t4), VapD expression decreased by 17% (0.832345) for below the basal value (t1). From 60 hours (t5) to 96 hours (t8) of intracellular *H. pylori* infection, VapD expression gradually increased from 1.2388716 to 2.262038 (by 126% higher than the reference value) (Fig 3). It is interesting to note that *H. pylori* VapD expression was increasing while it remained longer in the intracellular niche. A possible explanation for this is that *H. pylori* uses an increase in VapD expression as a protective mechanism against stress exerted by eukaryote cells, which also became under stress due to the depletion of nutrients from the cell culture medium. All values of *vapD* transcription levels of *H. pylori* were inversely proportional to the transcription levels values of GADPH, the latter was decreasing proportionally to the aging of the culture cells. The GADPH transcription levels increased only when the cell culture medium was

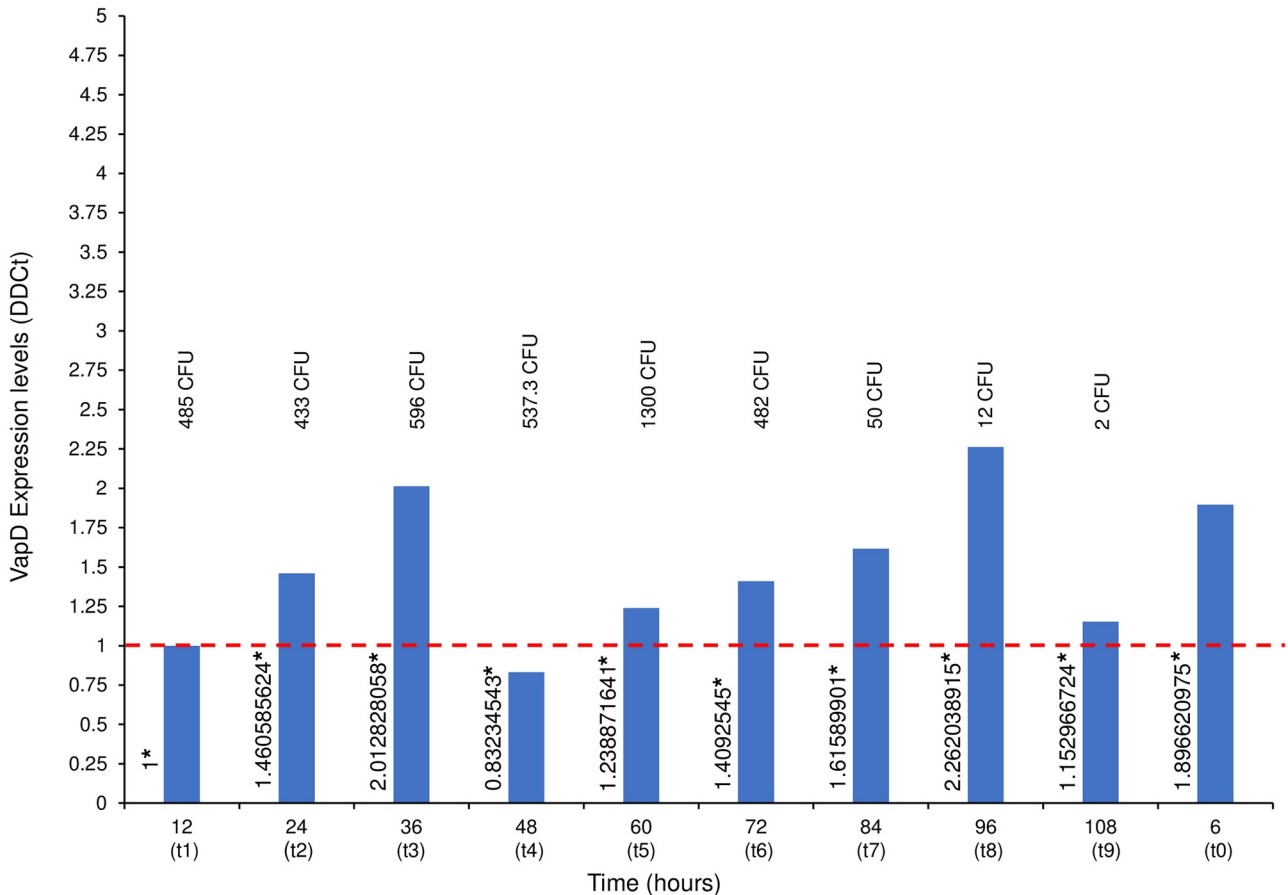

**Fig 3. VapD expression levels of *H. pylori* strain 26695 inside AGS cells.** RNA was extracted from co-culture (intracellular *H. pylori*- AGS cells) at different times: t0 (6 hours post-infection and pre-gentamycin treatment); t1 (12-h [hours]); t2 (24-h); t3 (36-h); t4 (48-h); t5 (60-h); t6 (72-h); t7 (84-h); t8 (96-h) and t9 (108-h) post-infection and post-gentamycin treatment. VapD expression values were determined using qRT-PCR. At each time point, the CFU (Colonies Forming Unit) count is also given.

changed by fresh medium (after of 36 hours). This observation suggests that when the AGS cells had consumed the nutrients from the culture medium, they became under stress and in turn exerted a greater stress on intracellular *H. pylori*, forcing to microorganism to increase the VapD expression levels to compensate the adverse events exerted by the intracellular niche. We also wanted to know if *H. pylori* VapD is expressed when the bacterium is grown in synthetic medium (extracellular niche). *H. pylori* strain 26695 was grown in LB medium and RNA was extracted. The results showed that no VapD expression levels were detected by qRT-PCT. These results support the idea that *vapD* is a gene that can be induced and is expressed when *H. pylori* comes into close contact with eukaryotic cells.

Given that several reports [20,21,23–27] have previously described transcription levels of different proteins or virulence factors in response to *H. pylori* infection *in vivo*, and based on our results, we wanted to investigate if VapD expression occurs in gastric biopsies (*in vivo*) taken from patients with different severe gastric pathologies. We examined a total of 82 gastric biopsies: 41 from the gastric antrum and 41 from the gastric corpus from 41 patients with different diagnoses of cancer, peptic ulcer, follicular gastritis and chronic atrophic gastritis. We detected VapD expression levels from the gastric antrum and corpus in all patients, indicating that *vapD*-positive *H. pylori* strains were colonizing both anatomic sites and that VapD is expressed *in vivo*. The VapD expression levels detected from the antrum and from the corpus were very similar, with no significant difference between the two regions of stomach (S1 Table). However, when VapD expression levels were compared between the different pathologies, we found significant VapD expression in biopsies from the gastric antrum of patients with follicular gastritis (p = 0.029). In addition, our results showed that the over 57 years age group presented the highest VapD expression levels (p = 0.018) in both gastric antrum and corpus biopsies. It is likely that infection in these older patients had been acquired at an early age, persisting for many years following the initial infection. The significant VapD expression in these older subjects suggests that VapD is necessary to support the persistence of *H. pylori* in gastric cells for a long period of time and to maintain the chronicity of infection.

## Discussion

In the present study, we found that *H. pylori* persisted in AGS cells maintaining an active metabolism for a long period of time, up to 108 hours. Although we observed vacuole formation after 24 hours post-infection in the AGS cell cytoplasm, *H. pylori* was not observed in the vacuoles, with the majority of the microorganisms being localized around the nucleus. Over time, coccoid forms became more evident, although the bacillary form persisted inside AGS cells, a phenomenon that was reflected in low CFU counts. However, the VapD expression did not change. *H. pylori* was mainly present in the gastric mucosa in a spiral-shaped form, but on aging the microorganism loses its typical spiral-shape and converts to a coccoid shape. Previous authors have referred to these forms as the C-shaped and U-shaped forms and they are the precursors to the coccoid form. These C-shaped and U-shaped forms of *H. pylori* are considered to be bacterial intermediates that favor survival transforming them into an inactive phase (dormancy) [28,29], due to adverse condition, as temperature or pH changes, use of antimicrobial drugs, prolonged fasting when cultivated, deficiency of nutrients, effects of antisecretory drugs, etc. [30]. In the dormant state, the bacteria lower their own metabolic activity, maintaining them for a long period of time without replicating. This phenomenon was first described in the 1940s, when Staphylococcal infections were seen to recur, even after extensive treatment with high doses of penicillin [31]. These types of bacteria are termed "persisters" and they have clinical implications on a number of infectious diseases, such as tuberculosis, syphilis, and typhoid fever, where the immune system proves ineffective, and a single surviving bacterium is capable

of restarting an infection [32]. In some microorganisms, a molecular mechanism that has been implicated in this type of persistence is the toxin-antitoxin (TA) module, which consists of pairs of genes usually located in the same operon, in which one gene acts as a toxin and the other cancels out the toxin effect. The result of toxin activation is the arrest of cell growth, and the fluctuation in toxin levels above and below the threshold, which results in the coexistence of dormant and growing cells [33,34]. Many pathogenic bacteria known to enter dormant states have a plethora of TA genes. Under stressful environmental conditions, such as nutrient limitation or oxidative stress, TA modules are activated. At present, six types of TA modules have been described, of which, type II is prevalent in the bacteria [33,34]. In nontypeable *Haemophilus influenzae* (NTHi), a common commensal of the upper respiratory tract and a significant cause of respiratory tract infection in humans, the presence of two TA modules *vapXD* and *vapBC* has been described. Double deletion mutants in both TA modules showed attenuation of NTHi persistence in a chinchilla model of otitis media and in primary human respiratory epithelial tissue. NTHi *vapD* has been shown to display ribonuclease activity *in vitro* and both the *vapBC*-1 and *vapXD* TA loci were shown to maintain NTHi survival and virulence [16]. On the other hand, *Rhodococcus equi vapD* was shown to be involved in protecting the microorganism against the respiratory burst inside the macrophage, and VapD was overexpressed when *R. equi* was grown in acid conditions. It is known that *R. equi* plasmid-cured mutant for *vapD* loses the ability to survive in macrophages and fails to induce pneumonia in foals [13,14]. However, several studies emphasize the importance of VapD type proteins in bacteria pathogenesis. In *H. pylori*, there are few studies relating to *vapD* (HP0315) and the virulence of this microorganism. Kwon et. al. [19] made the structural and biochemical characterization of VapD (HP0315) in *H. pylori* strain 26695 and reported that it has purine-specific endoribonuclease activity. Although *vapD* (HP0315) is arranged as an operon with HP0316, which was found to be an antitoxin-related protein, HP0315 is not a component of the TA system. The authors concluded that HP0315 might be an evolutionary intermediate, which does not belong to a TA system. Three TA modules have been identified in *H. pylori*: HP0892-HP0893; HP0894-HP0895 and Hp0968-HP0967 [35–37]. Some authors have considered HP0315-HP0316 to be a TA module [37] but more research is needed to clarify if HP0316 is the antitoxin that inhibited the action of HP0315, in order to determine if the *vapD* (HP0315) gene participates in the persistence of *H. pylori* in gastric cells, and to identify the role of *vapD* in the pathogenesis of severe gastric disease and/or in chronic infection. In a previous study carried out by our group, in which we determined the phylogenic relationship of *vapD* (HP0315) of *H. pylori* with other *vapD* genes present in microorganisms of different phyla, we found that *H. pylori vapD* is closely related to the *vapD* of *Rhodococcus equi* (manuscript sent for publication). The results of the latter study led us to believe that *vapD* of *H. pylori* could have a similar role to *vapD* of *Rhodococcus equi*, which protects microorganisms inside the eukaryotic cells [13,14]. Our results *in vivo* showed that *vapD* expression in *H. pylori* (strain 26695) within AGS cells was maintained throughout the time of the study, with levels of VapD expression increasing up to two-fold at 36 (t3) and 96 (t8) hours, in relation to our reference value at 12 (t1) hours, maintaining its persistence inside the AGS cells. We believe that this large increment of VapD expression was an induced protection response against hostile conditions inside the gastric cell, which was necessary for the bacterium survives.

We wanted to know if *vapD* of *H. pylori* strain 26695 grown in BAP *(in vitro)* was transcribed. Our results suggested that *vapD* is a gene capable of being induced and transcribed only when *H. pylori* comes into contact with gastric cells (human gastric mucosa and AGS cells) but it is not expressed when *H. pylori* is grown in culture medium, because it could not be necessary for its survival outside eukaryotic cell. These results agree with those previously reported by Graham et. al., [25], in which they showed that *vapD* was expressed by *H. pylori* in

response to interactions with mammalian gastric mucosa but was not expressed when the bacterium was grown *in vitro* [25].

The *vapD* gene is a strain-specific gene contributing to higher genetic diversity of *H. pylori*. Its frequency is variable from 38% to 61% among strains depending on the study population; for example, Cao and Cover [9] reported a frequency of 61.3% between the *H. pylori* strains isolated from an American population. We previously reported that *vapD* gene had a frequency of approximately 38% among *H. pylori* strains isolated from Mexican patients, however, this frequency was higher (52%) when the results were obtained from patients colonized with *vapD*-positive strains since the majority of them presented a mixed infection with *vapD*-positive and *vapD*-negative strains [38]. Contrary to those previously results reported, in the present study we found that all patients expressed VapD in their gastric biopsies (*in vivo*). These results could be explained by the type of patients studied in the present study, who had severe gastric pathologies associated with *H. pylori*, although, there was significance in the VapD expression levels in gastric antrum from patients with follicular gastritis. Our results suggest that VapD could be considered a virulence factor, which favors *H. pylori* permanence for a long-time, thereby contributing to the development of severe lesions. One hypothesis would be that VapD favors the survival of *H. pylori* within the gastric cell, so that the infection becomes chronic for months or years until it progresses in a severe gastric disease, probably triggering a carcinogenesis process in the gastric epithelial cell. Furthermore, we found that the highest VapD expression levels were in patients above 57 years of age, it is probable that these patients have carried the *H. pylori* infection for many years. We know, that in countries like ours, the *H. pylori* infection is frequently acquired in childhood and most of the time the disease will develop in adulthood. Early childhood infection typically results in an extended length of infection and likely affords more opportunities for oncogenic mutations [39,40]. Therefore, we believe that *vapD* presence in *H. pylori* strains could be a genetic marker for chronicity.

It is important to note, that in Mexico, mixed infection by different *H. pylori* genotypes is a common phenomenon, where different *H. pylori* strains colonize a single stomach [41–43]. Unfortunately, in the present work, we did not isolate individual colonies of *H. pylori* from gastric antrum neither from corpus, so that, we could not make a genotypic characterization of individual colonies. The frequency of *vapD*-positive and *vapD*-negative strains that colonized our patients in this study was unknown. However, the VapD expression levels detected in all the biopsies were high enough to believe that an important number of *vapD*-positive *H. pylori* strains were colonizing both gastric antrum and corpus in all patients. The impact of *H. pylori* virulence factors on the development of gastroduodenal diseases [3,23,24] is well documented. CagA is an oncoprotein, which is injected into host cells via a pilus structure called type IV secretion system (T4SS) encoded for *cag*-PAI [3,23,24]. VacA is a cytotoxin that induces vacuole formation in eukaryotic cells. Its activity is variable and is determined by changes in three regions of the *vacA* gene, leading to multiple alleles. Interestingly, the s1/i1/m1 type of *vacA* is often linked to *cagA*-positive strains [3,4,23,24], which are associated with a 4.8-fold risk of progression for precancerous lesions compared to those infected with *cagA*-negative/*vacA* s2/m2 strains. Therefore, none of the virulence markers can be considered an independent factor for disease outcome. In fact, when multiple virulence factors are present, the risk of severe clinical outcome is greater.

However, the wide distribution of *H. pylori* strains of different genotypes, in different regions of the world and in different ethnic groups, leads to different gastric disease outcome. It supports the need for new virulence markers to identify type I *H. pylori* strains, which have been associated more frequently with severe gastric pathologies. An intracellular phenotype confers advantage to the microorganism by protecting against host defense mechanisms and

against antimicrobial intervention, thereby favoring bacterial persistence and the chronicity of the infection. Preliminary mutational studies of the *vapD* gene in different microorganisms have shown that VapD participates in the persistence of microorganisms inside epithelial or phagocytic cells [13,14,16].

## Conclusion

*H. pylori vapD* presented high transcription levels inside AGS cells increasing these levels up to two-fold above the reference value. Transcription of *vapD* did not occur when *H. pylori* was grown in synthetic (*in vitro*) conditions, confirming that VapD was produced in response to interactions with eukaryotic cells. *H. pylori* acquired a coccoid form inside AGS cells, but maintained high VapD expression levels, indicating an active metabolism. We suggested that VapD provides a protective mechanism for the bacterium inside gastric cells, allowing it to remain in a viable non-cultivable state. *H. pylori* VapD was expressed in gastric biopsies from patients with severe gastric pathologies, such as chronic atrophic gastritis, follicular gastritis, peptic ulcers and gastric cancer. However, higher VapD expression levels were found in gastric antrum from patients with follicular gastritis and in older patients for above 57 years of age. VapD contributes to the persistence of *H. pylori* inside the gastric epithelial cell, protecting the microorganism against hostile intracellular environment, and favoring the chronicity of infection over a long period of time. We believe that the *vapD* presence in *H. pylori* strains could be a genetic marker of chronicity. Further studies are required to determine the molecular mechanism of action of VapD protein and its interaction with other proteins. To elucidate if *vapD* is part of a toxin-antitoxin module and which is its antitoxin.

## Supporting information

**S1 Table. VapD expression of *H. pylori* from antrum and corpus biopsies of patients with severe gastric pathologies.**
(XLSX)

**S1 Raw images.**
(PDF)

**S2 Raw images.**
(PDF)

**S3 Raw images.**
(PDF)

**S4 Raw images.**
(PDF)

**S5 Raw images.**
(PDF)

**S6 Raw images.**
(PDF)

**S7 Raw images.**
(PDF)

**S8 Raw images.**
(PDF)

## Acknowledgments

We thank Francisca Trujillo for helpful in the laboratory techniques.

## Author Contributions

**Conceptualization:** Rosario Morales-Espinosa, Alejandro Cravioto.

**Data curation:** Luis F. Mundo-Gallardo, Sergio Ayala.

**Formal analysis:** Luis-Roberto Serrano, Rigoberto Hernández-Castro, Alberto Gonzalez-Pedraza, Joaquín Manzo-Merino.

**Funding acquisition:** Rosario Morales-Espinosa.

**Investigation:** Gabriela Delgado, Elizabeth Castillo, Luis F. Mundo-Gallardo, Joaquín Manzo-Merino, Sergio Ayala.

**Methodology:** Gabriela Delgado, Luis-Roberto Serrano, Elizabeth Castillo, Carlos A. Santiago, Jose L. Mendez.

**Resources:** Sergio Ayala.

**Supervision:** Rosario Morales-Espinosa, Alejandro Cravioto.

**Validation:** Carlos A. Santiago.

**Writing – original draft:** Rosario Morales-Espinosa.

**Writing – review & editing:** Rosario Morales-Espinosa, Rigoberto Hernández-Castro, Alejandro Cravioto.

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
