## [Decision Letter · Decision Letter 0]

10 Dec 2019

PONE-D-19-29061

High expression of Helicobacter pylori VapD in both the intracellular environment and human gastric biopsies from patients with severe gastric pathologies

PLOS ONE

Dear Dr Morales-Espinosa,

Thank you for submitting your manuscript to PLOS ONE. After careful consideration, we feel that it has merit but does not fully meet PLOS ONE’s publication criteria as it currently stands. Therefore, we invite you to submit a revised version of the manuscript that addresses the points raised during the review process.

The study is novel and of interest but must be reviewed for the english before being published 

We would appreciate receiving your revised manuscript by Jan 24 2020 11:59PM. To enhance the reproducibility of your results, we recommend that if applicable you deposit your laboratory protocols in protocols.io, where a protocol can be assigned its own identifier (DOI) such that it can be cited independently in the future. For instructions see: http://journals.plos.org/plosone/s/submission-guidelines#loc-laboratory-protocols

We look forward to receiving your revised manuscript.

Kind regards,

Valli De Re, Ph.D.

Academic Editor

PLOS ONE

Journal Requirements:

1. We note that you have included the phrase “data not shown” in your manuscript. Unfortunately, this does not meet our data sharing requirements. PLOS does not permit references to inaccessible data. We require that authors provide all relevant data within the paper, Supporting Information files, or in an acceptable, public repository. Please add a citation to support this phrase or upload the data that corresponds with these findings to a stable repository (such as Figshare or Dryad) and provide and URLs, DOIs, or accession numbers that may be used to access these data. Or, if the data are not a core part of the research being presented in your study, we ask that you remove the phrase that refers to these data.

Reviewers' comments:

Reviewer's Responses to Questions

**Comments to the Author**

1. Is the manuscript technically sound, and do the data support the conclusions?

Reviewer #1: Yes

2. Has the statistical analysis been performed appropriately and rigorously? 

Reviewer #1: Yes

3. Have the authors made all data underlying the findings in their manuscript fully available?

Reviewer #1: Yes

4. Is the manuscript presented in an intelligible fashion and written in standard English?

Reviewer #1: No

5. Review Comments to the Author

Reviewer #1: In this manuscript, the authors observed the cytological features of H. pylori infection on AGS cells. This paper also showed the expression pattern of a small virulence-associated protein VapD of H. pylori in AGS cells and in the biopsies of patients with different gastric diseases by qRT-PCR, and found that the VapD is only expressed in the interaction of H.pyori with eukaryotic cells, further, expressed much higher in the older patients. The results are very interesting, which should be helpful to understand the long persistence of H. pylori in human stomach.

However, the unique issue before acceptation is the English editing. Here, I show some places that are necessary to improve their expression.

1. Title. “High expression of Helicobacter pylori VapD in both the intracellular environment and human gastric biopsies from patients with severe gastric pathologies”, suggested to be “High expression of Helicobacter pylori VapD in both intracellular environment and biopsies from gastric patients with severity”.

2. L30, “…intracellular H. pylori-AGS cells cultures at different time points and…”, suggested to be “…intracellular H. pylori-AGS cell cultures at different time points and…”

3. L33, “Our results show which vapD of H. pylori presented high transcription levels inside AGS cells that increased up to twofold…”, suggested to be “Our results show that vapD of H. pylori presented high transcription levels inside AGS cells, which increased up to twofold…”

4. L71, “…genetic diversity at both the gene and chromosomal level.” Changed to be “…genetic diversity at both the gene and chromosomal levels.”

5. L90, “fusion phagolysosome”, changed to “phagolysosome fusion”.

6. L92, “…clinical implications, in that it can lead to chronic…”, changed to “…clinical implications, which can lead to chronic…”

Here, I just show the first section of the manuscript. I suggest the authors to check carefully the manuscript again. Some places contain grammatical errors.

On the other hand, some expressions are better not to use assertive sentences because it is just a possibility. For example, L450, “…because it is not necessary for its survival outside eukaryotic cell”, better improving to be “…because it could not be necessary for its survival outside eukaryotic cell”.

6. PLOS authors have the option to publish the peer review history of their article (what does this mean?). If published, this will include your full peer review and any attached files.

Reviewer #1: No

---

## [Author Response · Author response to Decision Letter 0]

30 Jan 2020

1. We have eliminated from manuscript the phrase “data not shown” (line 309). It was substituted for the Figure 1, which show the results cited in the former paragraph.

Response to Reviewers

1. With respect to the article title

We are agree with the Review #1 about the article title, it is more convenient a title as: “ High expression of Helicobacter pylori VapD in both intracellular environment and biopsies from gastric patient with severity ”

2-6. We have reviewed the English grammar throughout the manuscript and the appropriate corrections have been made, which are indicated in red. Among them are those indicated by the reviewer on lines 30, 33, 71, 90, 92; which now correspond to lines 30, 32, 72, 91, 93 respectively.

On the other hand, we also change all assertive expressions used in the manuscript, instead of it, we "suggest" or "propose"

---

## [Editor Report · Decision Letter 1]

25 Feb 2020

High expression of Helicobacter pylori VapD in both the intracellular environment and biopsies from gastric patients with severity

PONE-D-19-29061R1

Dear Dr. Morales-Espinosa,

We are pleased to inform you that your manuscript has been judged scientifically suitable for publication and will be formally accepted for publication once it complies with all outstanding technical requirements.

With kind regards,

Valli De Re, Ph.D.

Academic Editor

PLOS ONE

---

## [Editor Report · Acceptance letter]

28 Feb 2020

PONE-D-19-29061R1 

High expression of *Helicobacter pylori* VapD in both the intracellular environment and biopsies from gastric patients with severity 

Dear Dr. Morales-Espinosa:

I am pleased to inform you that your manuscript has been deemed suitable for publication in PLOS ONE. Congratulations! Your manuscript is now with our production department. 

With kind regards,

on behalf of

Dr. Valli De Re 

Academic Editor

PLOS ONE